# Carbon Dioxide Tornado-Type Atmospheric-Pressure-Plasma-Jet-Processed rGO-SnO_2_ Nanocomposites for Symmetric Supercapacitors

**DOI:** 10.3390/ma14112777

**Published:** 2021-05-24

**Authors:** Jung-Hsien Chang, Song-Yu Chen, Yu-Lin Kuo, Chii-Rong Yang, Jian-Zhang Chen

**Affiliations:** 1Graduate Institute of Applied Mechanics, National Taiwan University, Taipei City 10617, Taiwan; r08543006@ntu.edu.tw; 2Advanced Research Center for Green Materials Science and Technology, National Taiwan University, Taipei City 10617, Taiwan; 3Department of Mechanical Engineering, National Taiwan University of Science and Technology, Taipei City 10607, Taiwan; D10803004@gapps.ntust.edu.tw; 4Department of Mechatronic Engineering, National Taiwan Normal University, Taipei City 10610, Taiwan; ycr@ntnu.edu.tw; 5Innovative Photonics Advanced Research Center (i-PARC), National Taiwan University, Taipei City 10617, Taiwan

**Keywords:** atmospheric-pressure plasma, carbon dioxide, reduced graphene oxide, tin oxide, supercapacitor, flexible electronics

## Abstract

Pastes containing reduced graphene oxide (rGO) and SnCl_2_ solution were screen printed on carbon cloth and then calcined using a CO_2_ tornado-type atmospheric-pressure plasma jet (APPJ). The tornado circulation of the plasma gas enhances the mixing of the reactive plasma species and thus ensures better reaction uniformity. Scanning electron microscopy (SEM), energy-dispersive spectroscopy (EDS), and X-ray photoelectron spectroscopy (XPS) were performed to characterize the synthesized rGO-SnO_2_ nanocomposites on carbon cloth. After CO_2_ tornado-type APPJ treatment, the pastes were converted into rGO-SnO_2_ nanocomposites for use as the active electrode materials of polyvinyl alcohol (PVA)-H_2_SO_4_ gel-electrolyte flexible supercapacitors (SCs). Various APPJ scanning times were tested to obtain SCs with optimized performance. With seven APPJ scans, the SC achieved the best areal capacitance of 37.17 mF/cm^2^ in Galvanostatic charging/discharging (GCD) and a capacitance retention rate of 84.2% after 10,000-cycle cyclic voltammetry (CV) tests. The capacitance contribution ratio, calculated as pseudocapacitance/electrical double layer capacitance (PC/EDLC), is ~50/50 as analyzed by the Trasatti method. GCD data were also analyzed to obtain Ragone plots; these indicated an energy density comparable to those of SCs processed using a fixed-point nitrogen APPJ in our previous study.

## 1. Introduction

A supercapacitor (SC) is a passive energy storage device that has a higher energy density than conventional capacitors and a higher power density than batteries [1]. Many aspects of SCs have been investigated, including their low-cost manufacturing process [2,3], flexibility [4], thermal stability [5], electrochemical stability [6], and high power density [7]. The selection and preparation of electrode active materials play critical roles in determining the SC performance [8]. SCs have two main types of charge storage mechanisms: (1) electric double-layer capacitance (EDLC) [9], in which charges are stored on the electrode–electrolyte interface, and (2) pseudocapacitance (PC) [10], which is based on the Faraday redox reaction. Typically, in a combination of carbon-based materials, metal oxides, and conductive polymers, carbon-based materials contribute the EDLC, and the multivalent metal oxides and conductive polymers contribute the PC [11,12].

Graphene is a two-dimensional carbon material that was first successfully prepared by Novoselov and Geim in 2004 [13]. Since graphene has high conductivity, carrier mobility, and specific surface area, it has been widely used in fuel cells [14], solar cells [15], oxygen evolution reactions [16], oxygen reduction reaction [17,18,19], and SCs [20]. However, in practical applications, graphene may be agglomerated and stacked, and therefore, the charge accumulation of the electric double layer could be limited [21,22]. To increase the capacitance further, graphene is often compounded with a metal oxide to combine the EDLC and the PC [23]. The metal oxides used commonly in SCs include MnO_2_, Co_3_O_4_, V_2_O_5_, SnO_2,_ and RuO_2_ [11,20,24,25]. In the present study, rGO and SnO_2_ were combined and processed using a scan-mode CO_2_ tornado-type atmospheric-pressure plasma jet (APPJ). SnO_2_ is cost effective and has good electrochemical properties [20,25]. Recently, many methods have been developed for synthesizing rGO-SnO_2_ including one-step synthesis [26], microwave-assisted synthesis [27], solvothermal synthesis [22], dynamic assembly [28], and sonochemical preparation [29].

Atmospheric pressure plasma (APP) can be processed on large scale in a regular-pressure environment without using a vacuum system. This generally reduces the cost. In contrast to a vacuum plasma system, an APP is relatively dusty and is therefore suitable for robust materials processing. Frequently used APP systems include arc, corona, dielectric barrier discharge (DBD) plasma, and APPJ [30,31,32]. An APP contains reactive plasma species with different self-sustaining temperatures. The properties of APPs can mainly be designed by the electrode configuration and the excitation power source. With various operating temperatures, various materials processes such as thin film deposition [33], etching [34], biomedical processing [35], and surface modification [36] can be designed.

CO_2_ is a common greenhouse gas on Earth. In this light, enabling the simultaneous reuse and decomposition of CO_2_ would be very beneficial. Therefore, many studies have investigated the recovery, degradation, conversion, and decomposition of CO_2_ [37,38,39,40]. Nonthermal plasma technology shows promise for the conversion of CO_2_ [41]. Further, many studies have used CO_2_ plasma systems to modify and oxidize materials [42,43,44]. CO_2_ DBD plasma has also been used for oxidizing carbon-based materials for use as the electrodes of SCs [45].

Previously, we have processed carbon nanotube (CNT)–SnO_2_ SCs using a fixed-point nitrogen APPJ [46,47]. However, the adhesion of SnO_2_ and CNTs was not good enough. Our subsequent study indicated that a higher SC specific capacitance value could be achieved by replacing 50 wt% of CNTs with reduced graphene oxides (rGOs) [8,48], possibly owing to the higher specific surface area of rGOs. Therefore, we used rGOs with SnO_2_ for the direct APPJ processing of rGO pastes containing SnCl_2_ solution, and this approach resulted in promising adhesion between SnO_2_ and rGOs [49]. In the present study, a scanning CO_2_ tornado-type APPJ was used to process screen-printed rGO pastes containing SnCl_2_ solution. The use of a CO_2_ tornado-type APPJ is beneficial for CO_2_ reuse and decomposition. The SCs fabricated by the CO_2_ tornado-type APPJ show comparable performance to those fabricated by the nitrogen fixed-point APPJ.

## 2. Experimental Sections

### 2.1. Preparation of rGO-SnCl_2_ Pastes for Screen Printing

The rGO-SnCl_2_ pastes were prepared by mixing rGOs (thickness: <5 nm, Golden Innovation Business, New Taipei City, Taiwan), SnCl_2_ (purity: 98%, anhydrous, Acros Organics, Geel, Belgium) solution, ethyl cellulose (#46070 and #46080, Sigma, Munich, Germany), ethanol (purity: 95%, Echo Chemical, Miaoli, Taiwan), and terpineol (anhydrous, Aldrich, Munich, Germany). The rGO was used as an active material to provide EDLC; SnCl_2_ solution was used as a precursor for converting to SnO_2_; ethyl cellulose was used as a binder, and the ethanol and terpineol were used as solvents. The details are described elsewhere [49].

### 2.2. Fabrication of rGO-SnO_2_ Electrodes

Figure 1a–c shows the fabrication process of the rGO-SnO_2_ electrode. First, rGO-SnCl_2_ pastes were screen printed on a carbon cloth current collector three times. The printed area was 1.5 × 2 cm^2^, as shown in Figure 2b. Next, the rGO-SnCl_2_ pastes printed on the carbon cloth were calcined at 100 °C for 10 min to remove excess solvent. Then, the screen-printed electrodes were scanned one, three, five, seven, and nine times using CO_2_ tornado-type APPJ (AC-PG-E-02, Click Sunshine Co., Ltd., New Taipei City, Taiwan). Figure 2a shows the schematic diagram of the APPJ. The APPJ parameters are as follows: CO_2_ flow rate is 35 slm, the distance between the plasma jet and the substrate is 4 mm, the power is 700 W, and the frequency of voltage source is 33 kHz. Figure 2c shows the APPJ scanning routine. Owing to the use of a rotating jet (i.e., tornado-type APPJ), the plasma was formed at the circumference of the jet rather than at the center of the jet. To ensure a homogenous process, the scanning area was much larger than the screen-printed area. The time required for each scan was ~55 s. The distance between each horizontal scan was 2 mm, and the distance between the exit of the plasma jet and the substrate was 4 mm.

### 2.3. Preparation of Gel–Electrolyte and Assembly of rGO-SnO_2_ Supercapacitor

The polyvinyl alcohol (PVA)–H_2_SO_4_ gel–electrolyte was prepared by mixing 1.7 g of PVA (99+% hydrolyzed) and 1 M of H_2_SO_4_ (purity: 95–97%, AUECC, Kaohsiung, Taiwan) under magnetic stirring with a rotation speed of 300 rpm at 80 °C until the solution became clear.

Figure 1d−f shows the assembly procedure and the structure of SCs. First, the rGO-SnO_2_ electrode was attached to a PVC substrate. Then, 0.5 mL PVA-H_2_SO_4_ electrolyte was dropped uniformly on the rGO-SnO_2_ electrode and dried for one day. This step was repeated three times. Finally, two pieces of PVA-H_2_SO_4_ electrolyte-coated electrodes were sandwiched on the gel-electrolyte sides to form a sandwich-structured SC.

### 2.4. Characterization of Materials and SCs

A k-type thermocouple and module (NI-9211, National Instruments, Austin, TX, USA) were used to measure the sample temperature during APPJ processing. A spectrometer (Mars HS2000+, GIE Optics, Taipei, Taiwan) was used to detect the optical emission spectra (OES). The integration time for the OES detection was 16 ms. The water contact angle of the electrodes was measured using an optical goniometer (Model 100SB, Sindetake, Taipei, Taiwan). The morphology of the rGO-SnO_2_ nanocomposites was observed by scanning electron microscopy (SEM, JSM-7800 Prime, JEOL, Tokyo, Japan) with energy-dispersive spectroscopy (EDS) at magnifications of 3000×, 30,000×, 100,000×. The bonding configuration was characterized using X-ray photoelectron spectroscopy (XPS, Sigma Probe, Thermo VG Scientific, Waltham, MA, USA) with an Al-Kα source and EDS. Electrochemical measurements were performed using cyclic voltammetry (CV) and galvanostatic charging/discharging (GCD) with a two-electrode configuration. The CV measurement was performed using an electrochemical workstation (Zennium, Zahner-Elektrik, Kronach, Germany) with potential scan rates of 200, 20, and 2 mV/s and a potential window of 0–0.8 V. The GCD measurement was performed using another electrochemical workstation (PGSTAT204, Metrohm Autolab, Utrecht, The Netherlands) with a potential window of 0–0.8 V and charging/discharging currents of 4, 2, 1, 0.5, and 0.25 mA.

## 3. Results and Discussions

### 3.1. Basic Information about CO_2_ Tornado-Type APPJ

Figure 3a shows the substrate temperature during CO_2_ APPJ processing (one scan). The temperature is slightly higher than 50 °C before APPJ scanning because of the residual heat left by the warm-up scan. The temperature quickly increases to hundreds of degrees Celsius. Therefore, the APPJ is suitable for rapid thermal processing. The plasma directly bombards the substrate surface, and the instantaneous maximum temperature becomes ~350 °C. Temperature oscillations could be caused by the rotation of the jet (i.e., tornado-type APPJ). Figure 3b shows the OES of the CO_2_ APPJ. The peaks generated at wavelengths of 282.8, 297.9, and 519.7 nm are mainly caused by CO species [50,51]. The peak observed at 359.9 nm is caused by the transition (*v*_3_–*v*_16_) of CO_2_. The peak at 406.0 nm is caused by the excitation of O_2_. Further, atomic O lines can be found at 776.4 and 843.9 nm [50]. The emission peaks at 386.5 and 458.1 nm are assigned to the CN violet system; the emission peaks at 391.0, 425.5 nm are assigned to the tail bands of CN; and those at 463.7 and 469.0 nm, to Le Blanc’s system [52]. Balmer’s series of H_δ_, H_γ_, and H_α_ can be detected at 410.8, 433.5, and 657.0 nm, respectively [53]. Further, another emission peak of H_2_ can be found at 722.5 nm [54]. The reaction of the CO_2_ APPJ is quite complicated and involves reactions with environmental gases. Many characteristic peaks related to oxygen can be detected, indicating that the plasma system has a strong oxidizing property. CN species were generated owing to the use of a carbon source in CO_2_, which would react with nitrogen in the surroundings. In addition, the generation of hydrogen and some oxygen species was attributed to the splitting of water vapor in the atmospheric environments by Equation (1) [55,56].
2H_2_O → 2H_2_ + O_2_(1)

This phenomenon could have further increased the number of oxygen species. As a result, the oxygen species were produced by the splitting of both CO_2_ and water vapor.

### 3.2. Water Contact Angles of rGO-SnO_2_

Figure 4 shows the water contact angle of the rGO-SnO_2_ electrode. The as-deposited rGO-SnCl_2_ has a high-water contact angle of 120.9°. After CO_2_ APPJ processing, the water droplet penetrated into the carbon cloth instantly for all APPJ scanning times. The increased hydrophilicity is mainly due to the oxygen functional groups implanted by the APPJ. This is partly due to heat and partly due to the oxidizing nature of our CO_2_ APPJ. This shows that the CO_2_ APPJ treatment can also effectively improve hydrophilicity. The increased hydrophilicity can improve the contact between the electrode and the gel–electrolyte, thereby improving the capacitance values.

### 3.3. Surface Morphology of rGO-SnO_2_ Electrode

Appendix A shows SEM images of the rGO-SnO_2_ electrode at a magnification of 3000×. The surface of the as-deposited electrode is relatively smooth, and the rGOs cannot be identified clearly because too much ethyl cellulose is attached to their surface. This could degrade the EDLC provided by rGOs, resulting in poor SC performance. Upon increasing the number of APPJ scans, the graphene sheets or stacks become more obvious, and most of the ethyl cellulose is burned off.

Appendix A shows SEM images of the rGO-SnO_2_ electrodes at a magnification of 30,000×. A small number of SnO_2_ particles is seen in the sample with one APPJ scan. After more than five APPJ scans, these particles are uniformly distributed on the rGO sheets. In our previous study of rGO SCs, no nanoparticle could be seen [57]. This indicates the successful synthesis of metal oxides in the study. Figure 5 shows the SEM images of rGO-SnO_2_ electrodes at a magnification of 100,000×. SnO_2_ can be seen clearly; further, this SnO_2_ contributes PC to SCs.

### 3.4. Chemical Bonding Analyses by XPS

Figure 6 shows an XPS survey scan of the screen-printed electrode. Elemental identification was conducted based on the reference book [58]. The as-deposited electrode exhibits a little Si pollution. This Si may come from the screen-printing process. In addition, only C, O, Sn, and a little Cl can be detected. Table 1 shows the elemental ratio of C, O, Sn, and Cl. Higher C and lower Sn atomic contents are seen because a large amount of ethyl cellulose exists in the printed electrode. After CO_2_ APPJ processing, the C content apparently decreased, mainly owing to the removal of excess ethyl cellulose. Furthermore, the large increase in the O content indicates that the CO_2_ APPJ is fairly suitable for oxidizing carbon-based materials. The detectability of Sn increases after APPJ processing. Further, the Cl content decreases after APPJ processing. This demonstrates that the CO_2_ tornado-type APPJ can quickly decompose SnCl_2_ and convert it into SnO_2_.

Appendix A shows the EDS analysis of the as-deposited and APPJ-processed electrode (seven scans). In contrast to the detection depth of XPS, which is less than 10 nm, EDS can be used to obtain chemical information from great depths in samples [59]. Appendix Aa,b shows the EDS analysis of the as-deposited electrodes. The C content is 75.5% owing to the combined contribution from carbon cloth, rGOs, ethyl cellulose, and little organic solvent. The O content is contributed to by the ethyl cellulose, rGOs, and SnO_2_. The Sn and Cl signals arise from SnCl_2_. Appendix Ac,d shows the XPS results obtained when the electrode was scanned seven times using the APPJ. The differences in C and O contents are small between the as-deposited and the seven-times APPJ-scanned samples. After seven APPJ scans, the C content increased from 75.5% to 77.9%, whereas the O content decreased from 20.0% to 17.1%. The O content may have decreased owing to the burnout of ethyl cellulose. Further, the Cl content decreased from 2.2% to 0.5%, clearly indicating the decomposition of SnCl_2_.

### 3.5. CV Measurements

To confirm that the increased capacitance mainly arises from the deposited rGO-SnO_2_ and not from the carbon fibers of carbon cloth, bare carbon cloth SCs without rGO-SnO_2_ coatings were fabricated. The CO_2_ APPJ was also used for processing the carbon cloth before gel–electrolyte was spread on it. The areal capacitance is calculated using Equation (2) [23] as follows:(2)$$C_{A}=\frac{S}{\Delta V\times v\times A}$$
where *C_A_* is the areal capacitance (mF/cm^2^); *S*, the integral area of the total cyclic voltammetry loop; Δ*V*, the potential window (V); *v*, the potential scan rate (mV/s); and *A*, the area of each electrode. Appendix A shows the CV measurement of the bare carbon cloth SC processed using the CO_2_ APPJ. The cyclic curves are triangular owing to the EDLC of an ideal carbon SC [60]. Appendix A shows the relationship between the APPJ scan time and the areal capacitance value. The best performance can be obtained with seven APPJ scans. The improved areal capacitance mainly arises from the increased hydrophilicity. However, the capacitance is 4.48 mF/cm^2^, as evaluated under a potential scan rate of 2 mV/s; this value is much lower than that of rGO-SnO_2_ processed using the CO_2_ APPJ, as described below.

Figure 7 shows the CV measurement of the rGO-SnO_2_ SCs processed using the CO_2_ APPJ. Compared with the CV curves of the bare carbon cloth SC (Appendix A) and rGO SCs processed using the APPJ [61], the curve shapes are no longer squarish. This is attributable to the PC caused by the deposition of SnO_2_. A slight hump is detected at ~0.3 V in the forward scan (0 to 0.8 V) and ~0.2 V in the reverse scan (0.8 to 0 V); this small hump becomes clearer in the sample scanned seven times. Similar characteristics were noted in related studies [26,62]. These could be redox peaks caused by SnO_2_. The addition of SnO_2_ apparently provided PC with EDLC, thereby further increasing the capacitance value. With a slower potential scan rate, a larger areal capacitance can be obtained. This is because ions can have more reaction time to intercalate the surface materials, thereby providing PC. As the potential scan rate increases, the charges may not have sufficient time to undergo redox Faradaic reactions. Therefore, the PC effect becomes less obvious, leading to a smaller capacitance value [63]. Table 2 shows the relationship between the areal capacitance value and the number of APPJ scans. The rGO-SnO_2_ SC that was scanned seven times shows the optimal areal capacitance of 33.59 mF/cm^2^ under a potential scan rate of 2 mV/s. Based on our previous study with Raman spectra, an overlong plasma process will damage the structure of graphene. It will reduce the capacitance value [8]. As a result, the areal capacitance decreased with the electrode APPJ scanned nine times.

In addition, the current change with the scan rate in the CV curve follows the power–law in Equation (3) [5].
(3)$$i=av^{b}$$
where *i* is the response current at 0.3 V; *v* is potential scan rate; *a* and *b* are variable parameters. The *b*-values are calculated from *log*(*i*) vs. *log*(*v*). The electrodes with *b* = 1 are classified as EDLCs; the electrodes with 0.8 < *b* < 1 are corresponded to PCs; the electrode with 0.5 < *b* < 0.8 is attributed to battery-type behavior [64]. Appendix Ad and Figure 7d show the power–law relationship with carbon cloth and rGO-SnO_2_ SCs. In carbon cloth SCs, the *b*-values were 0.93~0.96, which is close to ideal EDLCs. In rGO-SnO_2_ SCs, the *b*-values were 0.86~0.9, reflecting the behavior of PCs.

### 3.6. Trasatti Analysis

The capacitance contribution ratio, calculated as EDLC/PC, can be easily determined by the Trasatti method [65]. The Trasatti method divides the charge storage mechanisms of the surface charge (C_out_), which is associated with EDLC, and the diffusion-controlled charge (C_in_), which is proportional to v^−1/2^ (v is the potential scan rate) and is associated with PC. The two mechanisms have different responses to different potential scan rates [66]. When the potential scan rate approaches zero, both C_in_ and C_out_ readily respond to applied electric fields. When the potential scan rate approaches infinity, only C_out_ responds to applied electric fields [67,68]. Appendix A shows the Trasatti plots and capacitive contribution of carbon cloth SCs. Appendix Aa,b shows the relationship between 1/*C_A_* (subscript *A* denotes the areal capacitance) and v^1/2^, which can be used to extrapolate the vertical axis (v^1/2^ = 0) to find the C_total_ = C_in_ + C_out_. Appendix Ac shows the relationship between *C_A_* and v^−1/2^, which can be used to extrapolate to the vertical axis (v^−1/2^ = 0) to find C_out_. C_in_ can then be calculated by subtracting C_out_ from C_total_. Appendix A lists the calculated capacitive contribution, indicating that most of the electrochemical reaction of the carbon cloth SCs is dominated by EDLC. This is reasonable because carbon fibers contribute capacitance to the EDLC mechanism. The capacitive contributions of rGO-SnO_2_ SCs are also calculated. Figure 8 shows the Trasatti plots of rGO-SnO_2_ SCs. Table 3 shows C_total_, C_in_, and C_out_, as well as the capacitive contribution of rGO-SnO_2_ SCs based on Trasatti analysis. The PC/EDLC ratio is ~50% for all APPJ scan times. This confirms that the rGO-SnO_2_ composites effectively improved the areal capacitance and provided both PC and EDLC. The improved PC mainly arises from SnO_2_ [25].

### 3.7. GCD Measurements

The GCD measurement was used to evaluate the areal capacitance. The areal capacitance is calculated using Equation (4) [23] as follows:(4)$$C_{A}=\frac{2I\times T}{A\times \Delta V}$$
where *C_A_* is the areal capacitance (mF/cm^2^); Δ*V*, the potential window (V); *I*, the constant discharging current (mA); *T*, the discharging time (s); and *A*, the area of each electrode. Appendix A and Appendix A show the GCD results of carbon cloth SCs processed using the CO_2_ APPJ. As the I–R drop would be large when the GCD current is too large, the GCD current was varied as 1, 0.5, 0.25, and 0.1 mA in this experiment. The charging/discharging curves are shaped like an isosceles triangle; this is characteristic of EDLC and confirmed the results of the CV results. The capacitance of the carbon cloth SC is much lower than that of the rGO-SnO_2_ SC, as described below. This suggests that the rGO-SnO_2_ coating processed using CO_2_ tornado-type APPJ significantly improves the capacitance value.

Figure 9 and Table 4 show the GCD results of rGO-SnO_2_ SCs. The GCD results show an optimal capacitance of 37.17 mF/cm^2^ with seven APPJ scans. The charging and discharging times are similar, indicating excellent Coulombic efficiency [69]. Finally, Appendix Ae and Figure 9f show the areal capacitance of the carbon cloth and rGO-SnO_2_ SCs under different charging/discharging currents. A smaller GCD charging/discharging current results in a larger areal capacitance owing to the response of the PC effect.

### 3.8. Ragone Plots

Ragone plots are used to evaluate the energy and power density of the SCs. There are, respectively, calculated with Equations (5) and (6) [70] as follows:(5)$$E_{A}=\frac{C_{A}\times \Delta V^{2}}{7.2}$$
(6)$$P_{A}=\frac{3.6\times E_{A}}{T}$$
where *E_A_* is the energy density (per unit area) (µWh/cm^2^); *C_A_*, the areal capacitance calculated by the GCD method (mF/cm^2^); Δ*V*, the potential window (V); *P_A_*, power density (per unit area) (mW/cm^2^); and *T*, the discharging time (s). Figure 10 shows the Ragone plots of the SCs. With screen-printed rGO-SnO_2_ and optimal CO_2_ APPJ processing, the rGO-SnO_2_ SC exhibits the best energy density of 3.304 µWh/cm^2^ under a discharging current of 0.25 mA and the best power density of 1.067 mW/cm^2^ under a discharging current of 4 mA. Compared with bare carbon cloth SCs, rGO-SnO_2_ apparently promotes energy density. In comparison to the rGO-SnO_2_ SCs processed using fixed-point nitrogen APPJ [49], the energy density of SCs processed by CO_2_ tornado-type APPJ is nearly comparable. The slightly lower energy density value could be partly due to the temperature difference. Previously, the working temperature of our nitrogen APPJ was stably maintained at ~500–600 °C; in contrast, the working temperature of the CO_2_ tornado-type APPJ was 200–350 °C. We did not increase the operating temperature of the CO_2_ tornado-type APPJ because the rotation and scanning modes of the plasma jet would make heat dissipate more easily forming the surface and thereby reducing the temperature.

### 3.9. Electrochemical and Mechanical Stability of SCs

The 10,000-cycle CV stability was tested under a potential scan rate of 200 mV/s. Figure 11a shows the results. The rGO-SnO_2_ SC has an areal capacitance of 18.84 mF/cm^2^ in the first cycle. After 10,000 cycles, its areal capacitance decreased to 15.86 mF/cm^2^. Therefore, the capacitance retention was 84.2%. Furthermore, the rGO-SnO_2_ SC was tested under repeated bending with a bending radius of 7.5 mm (1000 times). Figure 11b shows the results of the bending test; the areal capacitance increased by 7.7% after 1000 bending cycles. This is because a small bending mechanical stress mainly facilitates improved contact between the electrolyte and the rGO-SnO_2_ composites.

### 3.10. Illumination of LED by Charged SCs

To verify whether the rGO-SnO_2_ SCs fabricated using CO_2_ tornado-type APPJ can actually be used as a power source for external electronic components, we use charged SCs to power up an LED. Toward this end, two rGO-SnO_2_ SCs were connected serially and charged by a 4.5 V DC power source for only 10 s. Then, the charged SCs were connected to the LED. The LED can be lighted at least for 30 s by the charged SCs, as shown in Figure 11c and Appendix A.

## 4. Conclusions

A CO_2_ tornado-type APPJ was successfully used for converting rGO–SnCl_2_ pastes into rGO-SnO_2_ composites that were then used as the electrodes of flexible SCs. CO_2_ tornado-type APPJ treatment could also improve the hydrophilicity of the electrode materials and facilitate contact between the gel–electrolyte and the electrodes. XPS and EDS validated the conversion of SnCl_2_ into SnO_2_. The best-performing SC, processed using seven CO_2_ tornado-type APPJ scans, exhibited an areal capacitance of 37.17 mF/cm^2^ in the GCD test. Comparing with the results of bare carbon cloth SCs confirms that the contribution of the capacitance was mainly from CO_2_ APPJ-processed rGO-SnO_2_. Trasatti analysis shows that the capacitance contribution ratio PC/EDLC is ~50/50 with PC enhanced by the inclusion of SnO_2_. The SC shows a capacitance retention rate of 84.2% after a 10,000-cycle CV stability test. Further, no degradation was observed after 1000 bending cycles with a bending radius of 7.5 mm.

## Figures and Tables

**Figure 1 materials-14-02777-f001:**
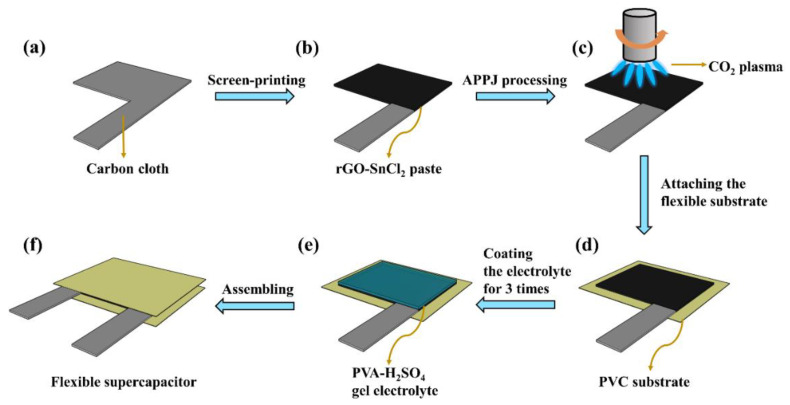
Schematic of fabrication process of rGO-SnO_2_ SC: (**a**) carbon cloth current collector; (**b**) screen printing; (**c**) processing by CO_2_ plasma; (**d**) attaching to flexible substrate; (**e**) coating gel–electrolyte; (**f**) assembling two electrodes to form sandwich-type SCs.

**Figure 2 materials-14-02777-f002:**
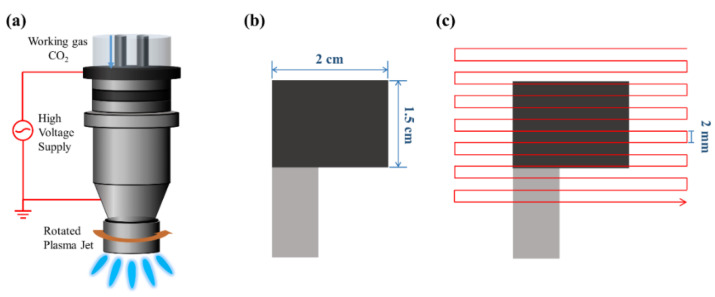
The schematic diagram of the APPJ processing: (**a**) schematic of CO_2_ APPJ system; (**b**) top view of screen-printed carbon cloth; (**c**) APPJ scanning routine.

**Figure 3 materials-14-02777-f003:**
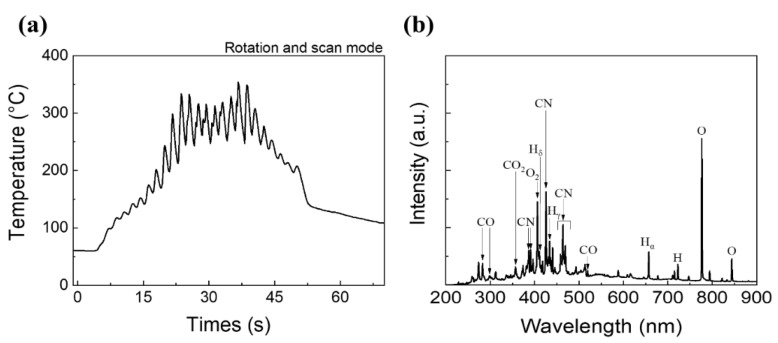
The basic information of CO_2_ APPJ: (**a**) temperature evolution of the substrate during APPJ scanning (one scan); (**b**) OES of CO_2_ APPJ.

**Figure 4 materials-14-02777-f004:**
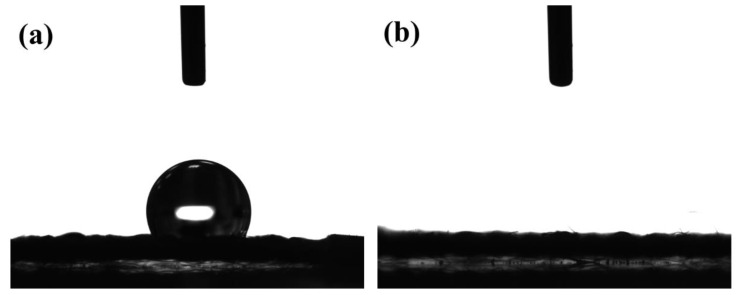
Water contact angle of screen-printed rGO–SnCl_2_ pastes scanned (**a**) zero and (**b**) APPJ-processed.

**Figure 5 materials-14-02777-f005:**
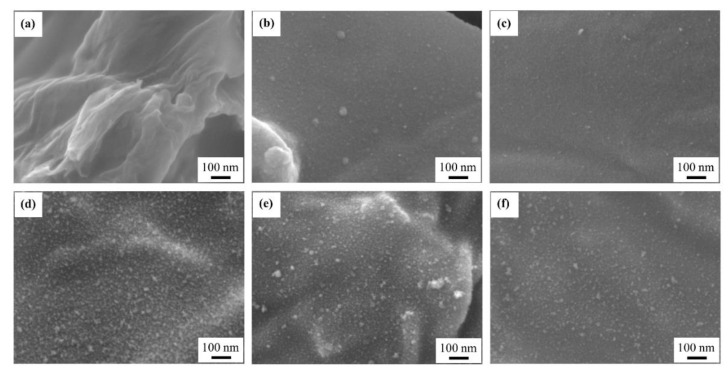
SEM images (100,000×) of rGO-SnO_2_ electrodes scanned (**a**) zero, (**b**) one, (**c**) three, (**d**) five, (**e**) seven, and (**f**) nine times using CO_2_ APPJ.

**Figure 6 materials-14-02777-f006:**
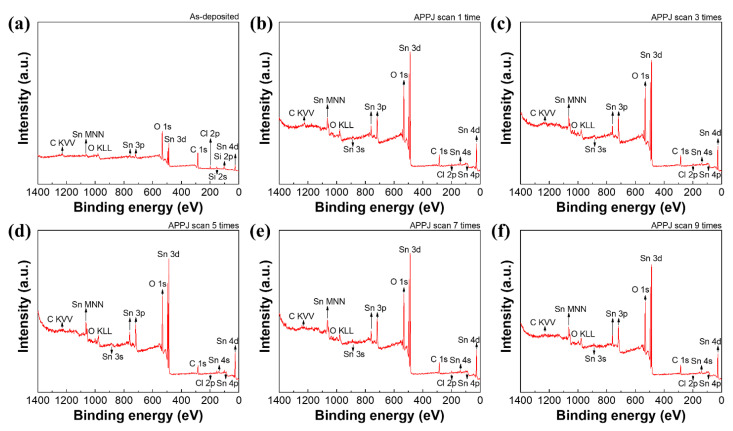
XPS analysis of electrodes scanned (**a**) zero, (**b**) one, (**c**) three, (**d**) five, (**e**) seven, (**f**) and nine times using the APPJ.

**Figure 7 materials-14-02777-f007:**
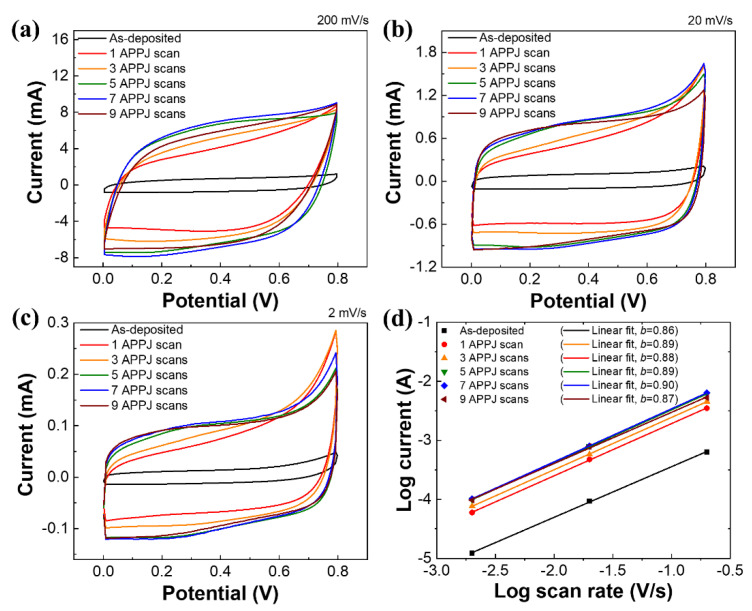
CV curves of rGO-SnO_2_ SCs under potential scan rates of (**a**) 200 mV, (**b**) 20 mV, and (**c**) 2 mV. (**d**) Logarithm of currents and scan rates.

**Figure 8 materials-14-02777-f008:**
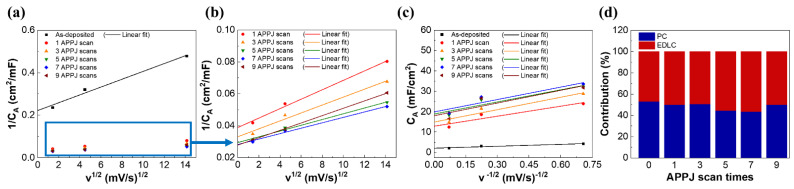
Trasatti plots of rGO-SnO_2_ SCs: (**a**,**b**) 1/C_A_ vs. v^1/2^; (**c**) C_A_ vs. v^−1/2^; (**d**) capacitive contribution of rGO-SnO_2_ SCs.

**Figure 9 materials-14-02777-f009:**
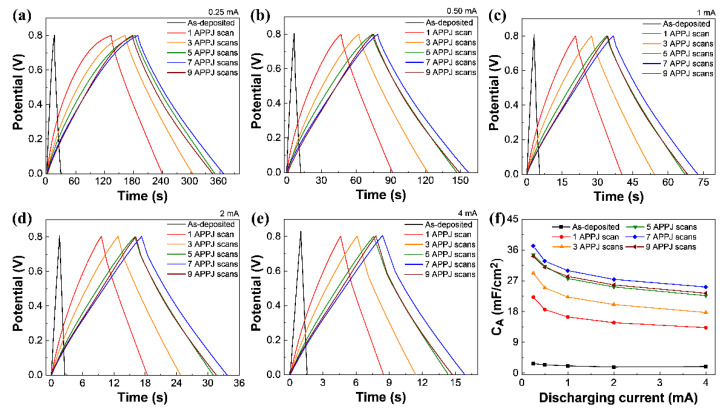
GCD of rGO-SnO_2_ supercapacitor under charge/discharge current of (**a**) 0.25 mA, (**b**) 0.50 mA, (**c**) 1 mA (**d**) 2 mA, and (**e**) 4 mA. (**f**) Areal capacitances calculated based on GCD results under different discharging currents.

**Figure 10 materials-14-02777-f010:**
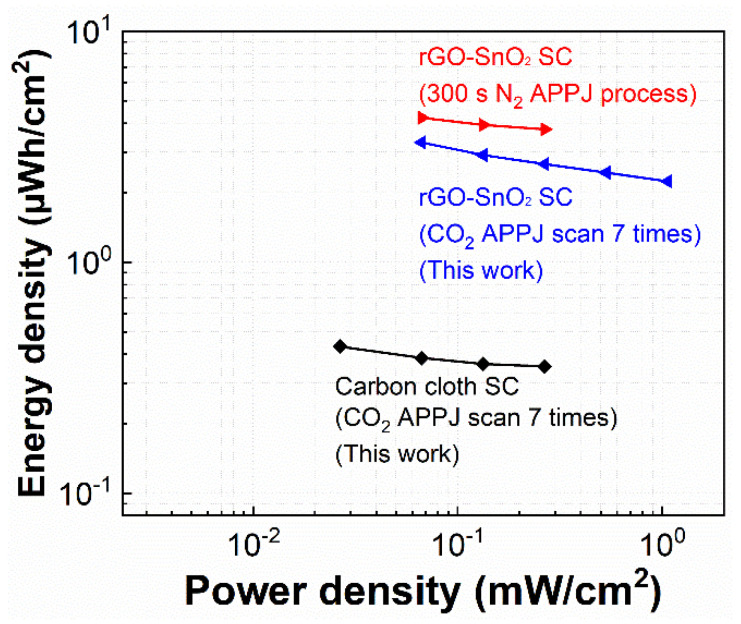
Ragone plots of rGO-SnO_2_. Comparison of results of CO_2_ tornado-type APPJ-processed rGO-SnO_2_ and bare carbon cloth SCs to those of fixed-point nitrogen APPJ-processed rGO-SnO_2_ SCs (our previous study [49]).

**Figure 11 materials-14-02777-f011:**
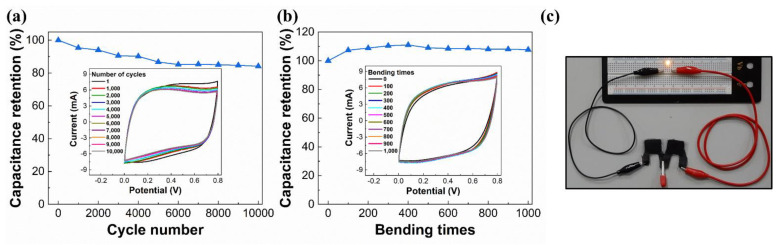
(**a**) CV cycling stability under a potential scan rate of 200 mV/s for 10,000 cycles; (**b**) cyclic bending test under a radius of 7.5 mm for 1000 cycles; (**c**) LED lit using two serially connected rGO-SnO_2_ SCs.

**Table 1 materials-14-02777-t001:** Elemental ratios analyzed based on XPS survey scan spectra in Figure 6.

Number of APPJ Scans	C (at.%)	O (at.%)	Sn (at.%)	Cl (at.%)
0	59.8	33.0	3.7	3.5
1	29.1	52.1	16.7	2.1
3	28.1	54.0	16.2	1.7
5	23.3	57.0	18.1	1.6
7	27.3	52.9	18.2	1.6
9	24.9	55.2	18.7	1.2

**Table 2 materials-14-02777-t002:** Areal capacitance of rGO-SnO_2_ SCs calculated based on CV curves shown in Figure 7.

Areal Capacitance (mF/cm^2^)			
Number of APPJ Scans	Potential Scan Rate (mV/s)		
	2	20	200
0	4.24	3.12	2.09
1	23.95	18.62	12.46
3	28.78	21.5	14.79
5	32.25	25.85	18.31
7	33.59	27.14	19.25
9	31.77	26.6	16.51

**Table 3 materials-14-02777-t003:** Capacitive contribution of rGO-SnO_2_ SCs.

Number of APPJ Scans	C_total_ (mF/cm^2^)	C_in_ (mF/cm^2^)	C_out_ (mF/cm^2^)	Capacitive Contribution (PC:EDLC) (%)
0	4.50	2.39	2.11	53.1:46.9
1	25.75	12.87	12.88	50.0:50.0
3	30.28	15.37	14.91	50.8:49.2
5	34.03	15.17	18.86	44.6:55.4
7	35.49	15.51	19.98	43.7:56.3
9	35.96	17.93	18.03	50.0:50.0

**Table 4 materials-14-02777-t004:** Areal capacitance of rGO-SnO_2_ SCs calculated based on GCD results.

Areal Capacitance (mF/cm^2^)					
Number of APPJ Scans	Charging/Discharging Current (mA)				
	0.25	0.50	1	2	4
0	2.75	2.40	2.10	1.75	1.89
1	22.19	18.58	16.41	14.73	13.28
3	29.21	24.93	22.22	20.05	17.71
5	34.57	31.23	27.64	25.19	22.63
7	37.17	32.74	29.96	27.40	25.17
9	34.73	30.92	28.27	25.78	23.36

## Data Availability

All data are included in the paper and the Appendix A.

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
