# Peer review of "Carbon Dioxide Tornado-Type Atmospheric-Pressure-Plasma-Jet-Processed rGO-SnO2 Nanocomposites for Symmetric Supercapacitors"

_materials, 2021, doi:10.3390/ma14112777_

Round 1
Reviewer 1 Report
The manuscript describes materials for symmetric supercapacitors. Pastes containing reduced graphene oxide (rGO)and SnCl2solution were screen-printed on carbon cloth and then calcined using a CO2 tornado-type atmospheric-pressure plasma jet (APPJ).
The manuscript presented concerns an interesting and actual subject. This manuscript can be accepted after major revision.
However, before publication the following major revisions should be taken into consideration below:
- In the introduction, (line 50-51) please add oxygen reduction reaction (ORR) and cite: (1) A review of oxygen reduction mechanisms for metal-free carbon-based electrocatalysts npj Comput Mater 5, 78 (2019). https://doi.org/10.1038/s41524-019-0210-3 (2) The Importance of Structural Factors for the Electrochemical Performance of Graphene/Carbon Nanotube/Melamine Powders towards the Catalytic Activity of Oxygen Reduction Reaction Materials 2021, 14(9), 2448; https://doi.org/10.3390/ma14092448 (3) The application of graphene and its composites in oxygen reduction electrocatalysis: a perspective and review of recent progress Energy Environ. Sci., 2016,9, 357-390 https://doi.org/10.1039/C5EE02474A.
- The overall English needs to be improved. Please seek guidance from a native English speaker if possible.
- Fig.5. The reference SEM image of rGO (raw) should be provided for comparison. Add some comments in the manuscript.
- Please explain the concentration of the C element. EDS and XPS method have a different ratio of C:O in these two methods, what authors say about that? Is EDS a good method to analyze these elements?
- Please add the discussion of the characterization by RAMAN spectroscopy.
- Please correct Figure 11a and 11b to better quality.
Thank you for the opportunity to review this manuscript.
Reviewer 2 Report
Please provide the thickness and mass of the rGO-SnO2 electrode after APPJ scanning.
In Figure 4 Please consider about remove c to f since there is no change in contract angle.
In Figure 5 please provide the reason why the number of SnO2 reduced after five times using Co2 APPJ or it is due to the SEM image quality.
In Figure S2 Please retake or change the contract of SEM figures to allow more clear show of SnO2 particles
In Page 8 the increase in area capacitance of carbon cloth was credited to the improved hydrophilicity as the number of Co2 APPJ process increase. Please provide the relationship between hydrophilicity level for carbon cloth and the number of Co2 APPJ scanning process.
In all CV and GC graphs please change the word “scan’ to APPJ scans as it can be easily mixed with CV scan
Please provide the reason for the area capacitance decreased with the electrode APPJ scanned for 9 times
Reviewer 3 Report
In the current manuscript “Carbon dioxide tornado-type atmospheric-pressure plasma jet processed rGO-SnO2 nanocomposites for symmetric supercapacitors” the authors studied the preparation of rGO-SnO2 by CO2 tornado-type atmospheric-pressure plasma jet. Structural and electrochemical characterization of electrodes on carbon cloth was done.
Combination of graphene or rGO with SnO2 is very popular and widely studied (not all of the mentioned in the manuscript but it is authors decision).
Authors are working in this area and already published the numbers of papers and 7 of them already cited in the current manuscript [Ref. 8,12,43,44,45,46,57].
However, there is missing information:
- There is no XRD of pure rGO, SnC2l and rGO-SnO2 as-deposited and after APPJ (to see the transformation of SnCl2 into SnO2).
- Author studied full-cell supercapacitor and mentioned about the capacitive behavior of device. However, there are no results for single electrode that needs to be present for the better understanding before the fabrication SC: do you have EDLC, pseudocapacitive or battery-type single electrode? It is important for the classification of the final device.
- Impedance is absent (Bode plot).
- Table for comparison data of obtained rGO-SnO2 with that prepared by other methods.
- For better comparison the values of gravimetric capacitance needs to be present also.
- Parameter b could be also calculated (b=1 means EDLC, b=0.5 means battery-type (easy to find in many paper, i.e. see https://doi.org/10.1021/acsnano.8b01914 or https://doi.org/10.3390/nano11051240)
- There is no explanation: why rGO-SnO2 prepared under CO2 in the current work has only 37 mF/cm2 in contract to 97 mF/cm2 reported for rGO-SnO2 prepared by the same authors but under N2 [Ref. 46]. What is the reason?
- Data for another rGO-SnO2 need to be added into Ragone plot.
- How long is 1 time of APPJ process? How it is comparable with 300s mentioned in Ragone plot for rGO-SnO2 prepared under N2?
Thus, based on the described before information and according to the instructions for Reviewers, my recommendation is Reject (article has serious flaws, additional experiments needed, research not conducted correctly).
Round 2
Reviewer 1 Report
Accept in current form
Author Response
We thank the reviewer for the valuable comments.
Reviewer 3 Report
From 9 points mentioned by me for corrections the authors applied only 2 (#6 and #9).
The authors
- didn’t show XRD (point 1) because of “produced nanoparticles have poor crystallinity”. It means authors studied amorphous layer of rGO-SnO2?
- didn’t analyze the single electrode (point 2), thus we need to believe that here is really supercapacitor made on rGO-SnO2 with pseudocapacitive behavior, not device with rGO-SnO2 such as here [Electrochemical Performance of SnO2 and SnO2/MWCNT/Graphene Composite Anodes for Li-Ion Batteries, http://przyrbwn.icm.edu.pl/APP/PDF/131/a131z1p57.pdf]
- didn’t show Bode plot (point 3) because “electrode impedance value was greatly affected by the measurement point connection, and the reproducibility of the transfer resistance data was not good”
- didn’t prepare Table for comparison (point 4) because they don’t have enough data for areal capacity and they don’t want to calculate gravimetric capacitance (point 7). The same reason is for Regone plot (point 9).
Thus, the current manuscript still is far from ideal and looks as report without comparison.
Are results good or bad?
At least it is possible to show both areal and gravimetric capacitance as well as both “energy density vs power density” and “gravimetric energy vs gravimetric power”.
Moreover, authors mentioned that “..We also have consulted with the industrial company fabricating supercapacitors. They also say the most important thing is the overall capacitance and stability...”
It is true, moreover, company is interesting about volumetric capacitance but authors cannot show the thickness as well as mass load.
Based on information before and position of authors I will choose “Accept after minor revision (corrections to minor methodological errors and text editing)”. Remove “This is a table.” In the title of the Table 1.
However, I hope that further papers of these authors will have higher quality than the current manuscript.
Author Response
Response) We thank the reviewer for the valuable comments. We have removed “
This is a table” in the title of Table 1.